# Overcrowded housing reduces COVID-19 mitigation measures and lowers emotional health among San Diego refugees from September to November of 2020

Ashkan Hassani[1,2☯]*, Vinton Omaleki[1,2☯], Jeanine Erikat[3], Elizabeth Frost[1,2,4], Samantha Streuli[1,2], Ramla Sahid[3], Homayra Yusufi[3], Rebecca Fielding-Miller[1,2,5]

1 Herbert Wertheim School of Public Health, University of California, San Diego, La Jolla, California, United States of America, 2 School of Medicine, Center on Gender Equity and Health, University of California, San Diego, La Jolla, California, United States of America, 3 Partnership for the Advancement of New Americans, San Diego, California, United States of America, 4 Joint Doctoral Program in Public Health (Global Health), San Diego State University, San Diego, California, United States of America, 5 School of Medicine, Division of Infectious Disease and Global Public Health, University of California, San Diego, La Jolla, California, United States of America

☯ These authors contributed equally to this work.
* ash055@ucsd.edu

## Abstract

Refugee communities are vulnerable to housing insecurity, which drives numerous health disparity outcomes in a historically marginalized population. The COVID-19 pandemic has only worsened the ongoing affordable housing crisis in the United States while continuing to highlight disparities in health outcomes across populations. We conducted interviewer-administered surveys with refugee and asylum seekers in San Diego County at the height of the COVID-19 pandemic to understand the social effects and drivers of COVID-19 in one of the largest refugee communities in the United States. Staff from a community-based refugee advocacy and research organization administered the surveys from September—November 2020. 544 respondents participated in the survey, which captured the diversity of the San Diego refugee community including East African (38%), Middle Eastern (35%), Afghan (17%), and Southeast Asian (11%) participants. Nearly two-thirds of respondents (65%) reported living in overcrowded conditions (> 1 individual per room) and 30% in severely crowded conditions (≥ 1.5 individuals per room). For each additional person per room, self-reported poor emotional health increased. Conversely, family size was associated with a lower likelihood of reporting poor emotional health. Crowded housing was significantly associated with a lower probability of accessing a COVID-19 diagnostic test, with every additional reported person per room there was approximately an 11% increase in the probability of having never accessed a COVID-19 testing. Access to affordable housing had the largest effect size and was associated with fewer people per room. Overcrowding housing is a structural burden that reduces COVID-19 risk mitigation behaviors. Improved access to affordable housing units or receiving vouchers could reduce overcrowded housing in vulnerable refugee communities.

**Data Availability Statement:** We have been given permission by our partner and owner of this data set, the Partnership for the Advancement of New Americans (PANA) to publicly store this data set in the University of California San Diego Library Research Data Curation program. Individuals interested in accessing the data may request the data set from library.ucsd.edu/dc under [https://doi.org/10.6075/J0PK0G9B].

**Funding:** Funding for this research was provided by the Partnership for the Advancement of New Americans (PANA). Additional support for analyses was provided by a UCSD Dissemination and Implementation Science Core pilot grant and three grants from the National Institutes of Health (K01MH112436, U01HD108787, T32AI007384). Additionally, our partners at PANA released a comprehensive report with further recommendations around housing. Authors JE, RS, and HY are PANA staff. The funders took a role in the study design and data collection, and review of the manuscript.

**Competing interests:** The authors have declared that no competing interests exist.

# Introduction

## Background

The immigrant and refugee community in the United States has experienced high rates of morbidity and mortality as a result of the COVID-19 pandemic [1, 2]. Globally, in high-income countries, foreign-born residents are at higher risk of COVID-19 transmission, have lower access to testing, and have higher rates of hospitalization and mortality due to COVID-19 compared to native-born residents [2]. In San Diego County, one analysis of death certificates found that although foreign-born residents comprise 23% of county residents, they accounted for 40% of all deaths attributed to COVID-19 between March 22, 2020 and March 22, 2021 [1]. Increased COVID-19 vulnerability in the immigrant and refugee community stems from a number of social and structural factors: Recent immigrants are more likely to work low-wage, frontline jobs, which carry a higher risk of disease acquisition and transmission, and are less likely to provide paid sick leave [3, 4]. Concerns about the public charge and a lack of documentation on immigration status may also deter individuals from seeking timely care when an infection is suspected [5, 6]. As a consequence of working low-wage jobs and racially discriminatory housing practices, immigrants are 4 times more likely to live in crowded housing [7]. This can exacerbate secondary attack rates within household units and present challenges for safe isolation [7].

Overcrowding in homes—sometimes referred to as 'hidden homelessness' has been highlighted by activists and researchers as a major issue in the refugee community in the United States [8–10]. Data from Canada and Europe document the increased risk of infectious disease among refugee communities living in overcrowded housing [8, 11]. Access to affordable housing (defined by the Department of Housing and Urban Development as 'housing that costs no more than 30% of your income) is associated with benefits such as improved food security, health care access, reduced stress, housing stability, improved indoor environmental health, and increased mental health [12]. Despite these benefits, access to affordable housing is severely limited by a backlogged federal Section 8 voucher program for affordable housing and state and local policies that have restricted new constructions of affordable housing and banned public housing [13].

## Objectives

Immigrant and refugee communities in the United States have been raising the issue of housing as a health and human rights issue for many years [10, 14]. As part of their biennial community survey conducted during the first peak of the pandemic on the state of the refugee community in San Diego County, the Partnership for the Advancement of New Americans (PANA) partnered with the University of California, San Diego (UCSD) to better understand the link between housing and COVID-19 vulnerability among refugee communities in Southern California.

## Methods

### Study design and setting

The Partnership for the Advancement of New Americans (PANA) is a research, public policy, and community organizing hub that serves refugees and asylum seekers in San Diego County [10]. Data for the present cross-sectional study were collected between September 2020 and November 2020 as part of PANA's biannual community survey. The survey was designed in English and translated into Arabic and Spanish by bilingual study staff. Interviewers were provided with a list of PANA members and the survey was subsequently administered over the

phone and in person by trained research assistants in Arabic, Burmese, Dari, English, Karenni, Oromo, Pashto, Somali, Spanish, and Swahili. Up to two household members could be interviewed. However, interviewers tracked whether an individual had completed a survey or not. Data were collected as part of a programmatic report intended to provide a general overview of the community. As such, we did not conduct *a priori* power calculations focused on one specific outcome.

Interviewers entered participant answers using Qualtrics software (Qualtrics ver. 2021). The survey contained 83 questions regarding demographics, COVID-19, housing, employment, health, children, belonging, and resilience. The survey took approximately 30 minutes to complete and up to 1 hour if it was being translated. The study team met weekly to discuss any issues with translation and address questions raised by survey administrators. Consistent with our participatory action approach, all data collection and analyses were conducted in conversation with PANA staff and guided by their lived expertise and organizational policy priorities. Variables for the present analyses were chosen in consultation with PANA leadership to address the primary research question: What is the association between housing and COVID-19 vulnerability? Vulnerability was broadly understood as both healthcare access and the broader mental health impact of the epidemic [15]. The full report with all variables can be found elsewhere [10].

## Participants

Participants were recruited using a convenience sampling approach. Individuals were eligible to participate in the study if they were over the age of 18, could speak one of the languages included in the study, willing to provide informed consent, and part of the refugee community. "Refugee community member" was construed broadly and included individuals who had arrived as refugees, asylum seekers, or their American-born children. PANA staff contacted all individuals for whom the organization had provided services or with whom they had engaged in community organizing efforts in the previous 5 years. Each staff interviewer was assigned a list of people based on language. Research assistants also recruited participants in person in shopping areas that members of the refugee community typically frequent. Staff interviewed and conducted the survey by phone with each individual from the convenience sample—ensuring no duplication. Interviews were limited to no more than two individuals per household (typically a parent and an adult child). All participants who completed the survey were given a $20 gift card to thank them for their time and expertise.

## Measures and analyses

Within the survey, participants were asked about the total number of individuals living in their homes and the total number of rooms in the home. In accordance with the California Department of Public Health (CDPH), overcrowding was defined as more than one individual per room. Severe overcrowding was defined as 1.5 individuals per room [16]. For example, a traditional 2-bedroom apartment would have 4 distinct rooms, including the kitchen and living room, and it would require 6 inhabitants to be classified as "severe overcrowding." Other explanatory variables included access to affordable housing (Are you in an affordable housing unit, or receiving a section 8 voucher to help with the rent?), access to a diagnostic test for COVID-19 (Have you ever gotten a test for COVID-19?), and self-rated emotional health with 5 potential responses ranging from "never" to "always" (How often have you been bothered by emotional problems such as feeling anxious, depressed or irritable?). In consultation with PANA leadership, the study team decided to dichotomize the emotional health variable for analysis and compare participants who reported 'never' experiencing emotional problems vs

all others. variables such as income from the previous week (Have you worked for money in the previous week), age, year of arrival to the United States, gender (male or female options only), cohabiting with a partner, number of children (How many children under the age of 18 are you responsible for?), and family size.

We first examined basic univariate frequencies as a study team to develop a basic understanding of sample demographics and the prevalence of indicators of interest within the sample. We then used chi-square and student's t-tests to test the hypothesis that our primary predictor and covariates of interest were significantly associated with living in severely overcrowded housing. We built simple logistic regression models to measure the unadjusted odds of association with overcrowded housing. Descriptive and bivariate statistical analyses were conducted using Stata 16 [17].

After assessing bivariate associates for both statistical and theoretical significance (the latter based on both the literature and discussions with the PANA leadership team) we constructed a structural equation model based on our hypothesized pathways between crowded housing, access to affordable housing support, reported emotional health, and the likelihood of accessing a COVID-19 test in the summer and fall of 2020. The model was fit to the data with the *Mplus* software package using a weighted least squares (WSMLV) estimator with probit link function to account for the mixture of binary, ordinal, and continuous variables [18]. Model fit was assessed based on individual covariate statistical significance along with global fit indices as recommended by Kline [19].

### Ethical considerations and IRB

Informed Consent was verbally obtained from all participants by trained PANA staff before administering the Qualtrics survey. Data gathered from the questionnaires was confidential. This study was reviewed and approved by the University of California, San Diego Institutional Review Board (UCSD IRB) under project #201601SX. In consideration of the vulnerable status of our study population, we adopted a participatory action approach to this community-led project and our team from UCSD only provided technical assistance [20].

## Results

PANA staff contacted 680 community members and 544 agreed to participate in the survey, for a response rate of 80%. The mean time living within the United States was 12 years, and gender was approximately evenly distributed (Table 1).

Just over 1 in 4 participants reported that they had engaged in work for money in the previous two weeks. Living in crowded or severely crowded housing conditions was common: participants reported an average of 1.5 individuals per room, with 29.9% (n = 160) living in severely overcrowded conditions. Thirty-two percent (n = 172) lived in affordable housing units or utilized Section 8 vouchers. At the time the study was conducted (September—November 2020), 23% of participants had ever accessed a diagnostic test for COVID-19 (n = 123). Approximately 37% of participants reported that they sometimes, half the time, most of the time, or always experienced emotional problems like anxiety or depression (Table 2).

The full structural model is shown in Fig 1. Global model fit statistics and structural path coefficients are shown in Table 3. Global fit statistics suggest that the model fit the data well, with a non-significant chi-square test (p = 0.22), Root Mean Square Error of Approximation below 0.05 (RMSEA = 0.03), Standardized Root Mean Square Residual below 0.1 (SRMR = 0.05), and Bentler Comparative Fit Index above 0.95 (CFI = 0.98).

**Table 1. Sample demographics of the refugee participants in San Diego, CA, sampled between November 2020-December 2020.**

| REGION | East Africa | Syria | Afghanistan | Southeast Asia | Total |
|---|---|---|---|---|---|
| | n = 205 | n = 184 | n = 90 | n = 56 | N = 535 |
| **Age** | | | | | |
| Mean (sd) | 38.5 (16.5) | 42.5 (10.1) | 35.8 (10.3) | 37.0 (14.8) | 39.3 (13.7) |
| Range | 15.0–96.0 | 15.0–68.0 | 14.0–70.0 | 14.0–68.0 | 14.0–96.0 |
| **Years in the United States** | | | | | |
| Median | 17.0 | 4.0 | 4.0 | 7.0 | 5.0 |
| Range | 1.0–35.0 | 1.0–43.0 | 0.0–35.0 | 1.0–26.0 | 0.0–43.0 |
| **English Spoken at Home** | 82 (40.0%) | 17 (9.2%) | 73 (81.1%) | 14 (25.0%) | 186 (34.8%) |
| **Female** | 150 (74.6%) | 38 (20.7%) | 11 (12.2%) | 41 (73.2%) | 240 (45.2%) |

For each additional person per room, the z-score of poor emotional health increases by 0.18, or approximately 0.18 points on the 1–5 likert scale. Conversely, family size was associated with a lower likelihood of reporting poor emotional health ($p < 0.001$). Each additional year in the United States was associated with a small, but statistically significant, lower probability of reporting poor emotional health ($p = 0.01$).

Crowded housing was significantly associated with a lower probability of accessing a COVID-19 diagnostic test in the summer and fall of 2020. With every additional reported person per room, the probability of having never accessed a COVID-19 test increased by 0.27 standard deviations, or approximately 11%.

The number of people per room, in turn, was significantly associated with access to affordable housing, reported family size, and years in the United States. Of these three, access to affordable housing had the largest effect size and was associated with a 0.37 standard deviation decrease, or 0.34 fewer people per room, with all other factors held constant ($p = 0.01$).

Female respondents were 6%, or 0.13 standard deviations, more likely to report having access to affordable housing ($p = 0.01$), as were individuals who had been in the United States

**Table 2. Severely overcrowded housing among participants in San Diego, CA from September to November 2020.**

| | Total (n = 535) | Not severely overcrowded (n = 375) | Severely overcrowded (n = 160) | |
|---|---|---|---|---|
| | *Mean (SD)* | *Mean (SD)* | *Mean (SD)* | p-value |
| **People Per Room** | 1.5 (0.9) | 1.1 (0.3) | 2.5 (1.0) | <0.001 |
| **Years in The United States** | 10.1 (8.5) | 12.0 (8.8) | 6.0 (6.1) | <0.001 |
| **Age** | 39.0 (13.8) | 39.8 (14.7) | 37.1 (11.2) | 0.040 |
| | *n (%)* | *n (%)* | *n (%)* | |
| **Never Accessed Covid-19 Test** | 410 (77.1) | 273 (73.2) | 137 (86.2) | 0.001 |
| **How often are you bothered by emotional problems like anxiety or depression?** | | | | 0.001 |
| Always | 21 (3.9) | 14 (3.7) | 7 (4.4) | |
| Most of the time | 19 (3.6) | 13 (3.5) | 6 (3.8) | |
| About half the time | 17 (3.2) | 15 (4.0) | 2 (1.3) | |
| Sometimes | 195 (36.7) | 99 (26.5) | 96 (60.8) | |
| Never | 280 (52.6) | 233 (62.3) | 47 (29.7) | |
| **Affordable Housing** | 172 (32.2) | 153 (40.9) | 19 (11.9) | <0.001 |
| **Employed** | 144 (27.2) | 95 (25.6) | 49 (31.0) | 0.200 |
| **Female** | 238 (44.9) | 189 (50.9) | 49 (30.8) | <0.001 |

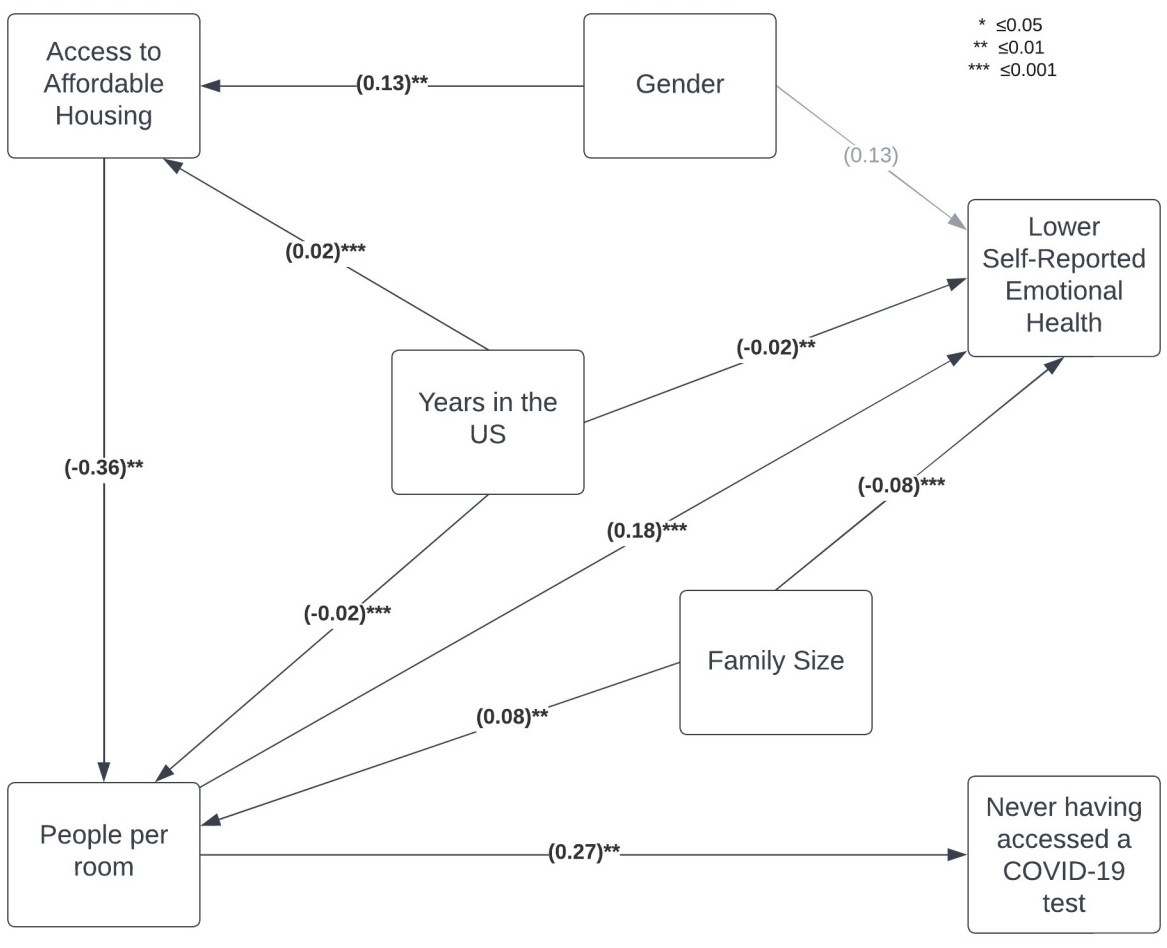

**Fig 1. Full structural model.**

longer, although each additional year was associated with only a small effect size, increasing just 1% per year (p<0.001).

## Discussion

### Key results

We found that within the refugee community, living in severely overcrowded housing was significantly associated with a decreased probability of accessing a COVID-19 test and an increased probability of reporting lower emotional health. Individuals who had never accessed a test for COVID-19 were more likely to live in severely crowded housing and significantly more likely to report lower emotional health. Meanwhile, those who reported accessing affordable housing were significantly less likely to report living in severely crowded conditions. Older age and more years lived in the United States were also associated with decreased odds of severely crowded housing conditions.

### Interpretation

The role of housing during the COVID-19 pandemic has focused mainly on how having shelter can mitigate risks of SAR-CoV-2 transmission and the need for moratoriums on evictions

**Table 3. Global structural model fit statistics and path coefficients.** B-coefficient depicts standardized change, or the number of standard deviations increased or decreased for each unit increase in the predictor variable. Unit change represents the b-coefficient value multiplied by the standard deviation of the outcome variable (i.e., people per room standard deviation = 0.98 multiplied by a standardized coefficient of 0.18 results in a 0.18-point increase [0.98 x 0.18] in reported poor emotional health).

| GLOBAL FIT | | | | |
|---|---|---|---|---|
| **Chi-square** | 9.42 | *df* | 7 | |
| | | p-value | 0.22 | |
| **RMSEA** | 0.03 | **CFI** | 0.98 | |
| **SRMR** | 0.05 | | | |
| **STRUCTURAL PATH COEFFICIENTS** | | | | |
| | *b-coefficient* | SE | p-value | Unit Change |
| **Poor Emotional Health** | | | | *(SD = 0.98)* |
| People per room | 0.18 | 0.04 | <0.001 | 0.18 |
| Female gender | 0.13 | 0.09 | 0.14 | 0.13 |
| Family size | -0.08 | 0.05 | 0.001 | -0.08 |
| Years in United States | -0.02 | 0.01 | 0.01 | -0.02 |
| **Never accessed covid-19 test** | | | | *(SD = 0.42)* |
| People per room | 0.27 | 0.09 | 0.002 | 0.11 |
| **People per room** | | | | *(SD = 0.91)* |
| Family size | 0.08 | 0.02 | <0.001 | 0.01 |
| Accessed affordable housing | -0.37 | 0.14 | 0.01 | -0.34 |
| Years in the United States | -0.02 | 0.01 | 0.001 | -0.02 |
| **Accessed affordable housing** | | | | *(SD = 0.47)* |
| Female | 0.13 | 0.05 | 0.01 | 0.06 |
| Years in United States | 0.02 | 0.01 | <0.001 | 0.01 |

[21, 22]. Overcrowded housing is a known risk factor for contracting and spreading COVID-19 [23–29], and the quality of housing and the density of inhabitants per domicile during the pandemic has garnered less attention, especially when regarding public health policy [20]. However, our work identifies a key mechanism through which overcrowded housing may also inhibit risk mitigation behaviors such as accessing COVID-19 diagnostic testing. Overcrowded housing may serve as an inhibition to testing: If an individual does not have the ability to safely isolate themselves from the rest of the household, then COVID-19 tests as tools to trigger isolation or quarantine behaviors become less salient. Given that access to therapeutic treatment for COVID-19 remains low, with lagging uptake in racial and ethnic minorities in particular [30, 31], these findings suggest that messaging which emphasizes the role of testing as a first step to accessing treatment, rather than a tool to trigger isolation, may be especially important to address ongoing disparities in COVID-19 morbidity and mortality.

We found a significant, meaningful association in the link between lower emotional/mental health and crowded housing in the refugee community. Discussing mental health is highly stigmatized in this community, and the study team agreed that disclosing any concerns related to emotional or mental health in this context was likely indicative of significant levels of emotional distress. There is evidence that overcrowded housing can have a negative impact on an individual's mental health in the general population [32, 33]. Our findings add to this literature but our narrowed focus on the refugee community highlights that housing conditions can add an extra emotional and mental burden to a population already at high risk of lower emotional/mental health outcomes [34, 35]. This combination of the mental strains that refugees in overcrowded housing experience may be alleviated with tailored mental health interventions [36].

Affordable housing significantly reduces the odds of living in severely crowded conditions within our study sample. There is an average wait of 8–10 years for applicants to receive a federal section 8 housing voucher to move into an affordable home which is consistent with our data that shows newly arrived refugees are at even higher odds of overcrowding, and that the longer they live in the United States, the more likely they are to live in affordable housing [37]. Our findings also suggest that refugees living in San Diego are experiencing economic factors that force multiple refugee families to share spaces meant for only one family. This is a form of 'hidden homelessness [11]. Just over 1 in 4 individuals reported engaging in work for money in the previous 2 weeks. While some of this number may be an artifact of sampling bias (i.e., those who responded were more likely to be free during the day to participate in the survey) and/or represent individuals who are full-time students or homemakers, previous work conducted by our team demonstrated that the refugee community experienced high rates of job loss during the pandemic [32]. These overcrowded housing arrangements may be necessary for economic survival, but in the context of the COVID-19 pandemic, they become hot spots of risk, increasing the likelihood of acquiring, transmitting, and reinfection with the virus [38].

Support for affordable housing access as a fundamental human right has grown globally [39]. The movement to decommodify housing by increasing social housing construction in proportion to privately funded housing has shown promise in tackling the global housing crisis [40, 41]. Cities like Tokyo or Vienna in particular have successfully handled growing housing demand by creating holistic policies that include social housing construction, limited-profit housing associations, and tenant protections [40, 42, 43].

Additionally, housing and location are significant influencers on assimilation and social mobility [40, 44]. Creating affordable housing that is near physical and social structures has been shown to increase social inclusion and integration [40, 44]. Therefore, creating affordable housing in the right locations is an essential stepping stone for the resettlement of refugees.

## Limitations

This is a cross-sectional study with a convenience sample. As such we cannot make claims about the directionality of the associations we identified, nor should our data be interpreted as representing a true prevalence. However, a full census of the refugee population in the region —or country—does not exist, making random sampling impractical if not impossible. Additionally, gender was limited to two options (male or female) to preserve statistical significance and we are aware that this may not be a true representation of the diverse population. Survey data was self-reported so it is possible certain topics were biased towards more socially acceptable answers. Furthermore, although the survey was limited to two participants per household, the survey did not record how many participants belonged in the same household. Therefore, we were unable to adjust household size as a fixed effect. However, PANA interviewers reported that in practice it was unusual for two respondents from the same household to participate in the survey.

## Conclusions

Overcrowded housing creates structural burdens that may prevent refugees from taking risk mitigation behaviors like diagnostic testing regarding airborne diseases such as COVID-19. These housing conditions may also be impacting refugee emotional health, signaling the need for mental health resources and further community advocacy in these communities. Promptly increasing access to affordable housing for vulnerable refugee populations may also protect against the increased odds of COVID-19 associated with overcrowding and help mitigate future outbreaks. We recommend that local governments utilize federal resettlement funding

in collaboration with community-based organizations to develop and implement affordable housing plans for refugees upon arrival. This will ensure that recently resettled refugees are able to build long-term stability while improving population health in the United States' most marginalized communities. In addition, we recommend the federal government increase the number of Section 8 vouchers available so that wait times are reduced [34]. Finally, we recommend that the State of California create legislation and provide funding for permanent social housing based on successful housing policies from other parts of the world, that refugee populations can immediately access [10]. Future directions in research should look at the impact of master leasing, housing vouchers, and universal basic income on refugee housing security and general health outcomes.

## Author Contributions

**Conceptualization:** Ashkan Hassani, Vinton Omaleki, Rebecca Fielding-Miller.

**Data curation:** Ashkan Hassani, Vinton Omaleki, Jeanine Erikat, Elizabeth Frost, Ramla Sahid, Homayra Yusufi.

**Formal analysis:** Ashkan Hassani, Vinton Omaleki, Jeanine Erikat, Rebecca Fielding-Miller.

**Funding acquisition:** Rebecca Fielding-Miller.

**Investigation:** Ashkan Hassani, Vinton Omaleki, Elizabeth Frost, Ramla Sahid, Homayra Yusufi.

**Methodology:** Ashkan Hassani, Vinton Omaleki, Jeanine Erikat, Rebecca Fielding-Miller.

**Project administration:** Elizabeth Frost, Ramla Sahid, Homayra Yusufi.

**Supervision:** Rebecca Fielding-Miller.

**Writing – original draft:** Ashkan Hassani, Vinton Omaleki, Jeanine Erikat, Rebecca Fielding-Miller.

**Writing – review & editing:** Ashkan Hassani, Vinton Omaleki, Jeanine Erikat, Elizabeth Frost, Samantha Streuli, Ramla Sahid, Homayra Yusufi, Rebecca Fielding-Miller.

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
