## [Decision Letter · Decision Letter 0]

24 Aug 2022

PONE-D-22-20522Overcrowded Housing Reduces COVID-19 Mitigation Measures and Increases Poor Emotional Health Among San Diego RefugeesPLOS ONE

Dear Dr. Hassani,

Thank you for submitting your manuscript to PLOS ONE. After careful consideration, we feel that it has merit but does not fully meet PLOS ONE’s publication criteria as it currently stands. Therefore, we invite you to submit a revised version of the manuscript that addresses the points raised during the review process.

ACADEMIC EDITOR: Overall, a well-written manuscript that can be improved. Definitely, an important topic, but needs a lot more research methodological detail. Authors need to address all peer-reviewer comments and critiques thoroughly. Lastly, please report with Strobe Guidine, were were recommended by Reviewer #2.

We look forward to receiving your revised manuscript.

Kind regards,

Uğurcan Sayılı

Academic Editor

PLOS ONE

Journal Requirements:

"Funding for this research was provided by the Partnership for the Advancement of New Americans. Additional support for analyses was provided by a UCSD Dissemination and Implementation Science Core pilot grant and the National Institutes of Health (K01MH112436, U01HD108787). Additionally, our partners at PANA released a comprehensive report with further recommendations around housing and beyond. There are no conflicts of interest."

Please state what role the funders took in the study.  If the funders had no role, please state: ""The funders had no role in study design, data collection and analysis, decision to publish, or preparation of the manuscript.""  If this statement is not correct you must amend it as needed. 

5. Please amend your authorship list in your manuscript file to include author "Samantha Strueli".

Reviewers' comments:

Reviewer's Responses to Questions

**Comments to the Author**

1. Is the manuscript technically sound, and do the data support the conclusions?

Reviewer #1: Yes

Reviewer #2: Yes

Reviewer #3: Yes

2. Has the statistical analysis been performed appropriately and rigorously? 

Reviewer #1: Yes

Reviewer #2: Yes

Reviewer #3: Yes

3. Have the authors made all data underlying the findings in their manuscript fully available?

Reviewer #1: No

Reviewer #2: No

Reviewer #3: Yes

4. Is the manuscript presented in an intelligible fashion and written in standard English?

Reviewer #1: Yes

Reviewer #2: Yes

Reviewer #3: Yes

5. Review Comments to the Author

Reviewer #1: This is a carefully written, thoughtfully designed study. It adds nuance to our understanding of SDoH for newcomers. It contributes to a body of literature on mitigating future pandemics.

Major concern:

1. I believe the survey protocol allowed up to 2 respondents per household. Does this mean the ~500 respondents might represent ~250 households? If so, would this not require attention in the model, e.g., household as a fixed effect?

Minor questions:

2. Could the authors clarify the use of the term "refugee" to characterize survey respondents? Based on the description of PANA and community venue-based recruitment, it sounds like respondents might be refugees, asylum-seekers, or others who speak the same languages but have arrived in the US via family sponsorship, work, etc.

3. Could the authors add a comment regarding the low employment rate?

4. I appreciate the description of weekly meetings to address translation challenges. Could the authors offer a more detailed description, e.g., as an appendix? It would be helpful to know more about the language and cultural qualifications (formal or informal) of the Arabic and Spanish translators and the steps study team members might have taken to ensure equivalence, e.g., meetings to review the English versions, proof readers, etc. Additionally, it sounds as if many languages were sight-translated from an English version of the survey? How were those team members trained to ensure comprehension of the meaning and intention of the English survey items? How did they communicate mental health vocab or other challenging vocab where linguistic and community equivalent terms may be harder to find? Did they develop a glossary for key terms?

5. Is there a reason the authors elected not to share data on differences between language groups or include language in the adjusted model? I realize this is a nuanced decision. But communities who speak languages of lesser diffusion may face enhanced barriers to Section 8 applications.

Reviewer #2: This manuscript is on an important topic for refugee health. The analysis is meaningful and would be an important contribution to the literature. However, the manuscript needs substantial revisions throughout. Below please find some feedback.

• Title

o I suggest using more of an objective word like low instead of the subjective word poor in the title and throughout.

o I suggest adding the time period in the title as outcomes change throughout time.

• Abstract

o The total sample size is 544, but it reads as half the sample size in the abstract. I suggest re-writing this to make it clearer.

o I suggest replacing “fewer” odds with lower odds.

• Introduction

o I suggest adding bolded subheadings throughout the introduction to guide the reader.

o A reference is needed for the first sentence.

o There are a couple of run-on sentences at the end of the first paragraph.

o Some references are numbered and some are authors last name.

o Period missing at the end of the introduction section.

• Methods

o The STROBE guidelines should be used to make the methods section clear and consistent with previous literature. Sections should include setting, participants, variables, data source, and statistical methods. https://www.equator-network.org/wp-content/uploads/2015/10/STROBE_checklist_v4_combined.pdf

o “Interviews were limited to no more than two individuals per household (typically a parent and an adult child).” – How many interviews were completed with individuals from the same household? This should be mentioned as a limitation in the discussion.

o How many questions were on the survey? About how long did it take for participants to complete?

o The statistical methods are mentioned before describing the survey and variables, which should come first.

o “Data gathered from the questionnaires was anonymous” – How do you know if a participants completed the survey more than once? This could be mentioned in the limitations.

o Data is based on self-report, which should be described in the limitations.

o What options were given for gender? Were there responses available for: prefer not to respond, non-binary, or transgender?

• Results

o The first paragraph of the results jumps into a comparison without first describing the overall population of respondents. How old were the respondents for example? What regions or countries are respondents from? What percent of respondents are refugees versus asylum seekers? How many of the respondents are refugees/asylees versus children of refugees/asylees born in the US?

o An overall demographics table would be valuable as an initial table.

o This sentence describes methods and should be moved to the methods section: “In consultation with PANA leadership, the study team decided to dichotomize this variable and compare participants who reported ‘never’ experiencing emotional problems vs all others.”

o This sentence interprets the results and should be moved to the discussion section: “Discussing mental health is highly stigmatized in the community, and the study team agreed that disclosing any concerns related to emotional or mental health in this context was likely indicative of significant levels of emotional distress.”

• Discussion

o I suggest adding bolded subheadings throughout the discussion to guide the reader.

o I suggest re-wording the findings to describe how more support is needed for this population as opposed to it reading as if this population is not taking care of their health and putting others at risk.

o The second paragraph of the discussion should be moved to a limitations section at the end of the discussion before the conclusions.

o More of a description of the time period is needed related to COVID-19 and testing. For example, how accessible were COVID-19 testing at that time? When did the government send out home tests for free?

o This sentence implies causality instead of an association: “housing conditions can add an extra emotional and mental burden”

o United States and US are used. I suggest writing out in full each time.

o “San Diegan” – Perhaps re-word to refugees living in San Diego.

o A limitations section was missing.

o It would be great to see more specifics about next steps related to affordable housing mentioned in the conclusions. What are some specific steps that could be taken to address this issue? Have other counties, states, or countries successfully found affordable housing solutions?

• Tables and Figures

o Tables should include more comprehensive titles that answer who (refugee and asylum seekers), what, where (San Diego County, CA), and when (time period).

o I suggest consistency in the number of significant digits in the p values in the tables.

Reviewer #3: A very nice community-engaged applied research study addressing an extremely important topic. I suggest that they re-arrange Table 1 so that they report measures in the table in the same order in which they report them in the text.

6. PLOS authors have the option to publish the peer review history of their article (what does this mean?). If published, this will include your full peer review and any attached files.

Reviewer #1: No

Reviewer #2: No

Reviewer #3: No

---

## [Author Response · Author response to Decision Letter 0]

22 Nov 2022

Thank you all for your feedback and suggestions.

Reviewer #1: 

This is a carefully written, thoughtfully designed study. It adds nuance to our understanding of SDoH for newcomers. It contributes to a body of literature on mitigating future pandemics.

Major concern:

1. I believe the survey protocol allowed up to 2 respondents per household. Does this mean the ~500 respondents might represent ~250 households? If so, would this not require attention in the model, e.g., household as a fixed effect?

Response: This is an important observation and we agree with the reviewer’s critique. While up to 2 individuals were technically eligible per household, it was rare that more than one person actually participated in reality. In large part because it was rare, household-level data were not collected and so we are not able to include a mixed effect in the model. We have acknowledged this flaw in our data with the following text added to the limitations section:

Furthermore, although the survey was limited to two participants per household, the survey did not record how many participants belonged in the same household. Therefore, we were unable to adjust for a household size as a fixed effect. However, PANA interviewers reported that in practice it was extremely for two respondents from the same household to participate in the survey.

Minor questions:

2. Could the authors clarify the use of the term "refugee" to characterize survey respondents? Based on the description of PANA and community venue-based recruitment, it sounds like respondents might be refugees, asylum-seekers, or others who speak the same languages but have arrived in the US via family sponsorship, work, etc.

Response: All families interviewed are either recent arrivals or folks who have become new Americans and all arrived as refugees. Youth interviewed might have been born in the U.S. but their parents arrived as refugees. We have clarified by adding the text below:

“Refugee community member” was construed broadly and included individuals who had arrived as refugees or asylum seekers or their American born children”

3. Could the authors add a comment regarding the low employment rate?

Response: Thank you for this observation – we have expanded the discussion to include a brief discussion of what we agree is a surprisingly low employment rate. The following text has been added:

Just over 1 in 4 individuals reported engaging in work for money in the previous 2 weeks. While some of this number may be an artifact of sampling bias (i.e., those who responded were more likely to be free during the day to participate in the survey) and/or represent individuals who are full time students or homemakers, previous work conducted by our team demonstrated that the refugee community experienced extremely high rates of job loss during the pandemic

4. I appreciate the description of weekly meetings to address translation challenges. Could the authors offer a more detailed description, e.g., as an appendix? It would be helpful to know more about the language and cultural qualifications (formal or informal) of the Arabic and Spanish translators and the steps study team members might have taken to ensure equivalence, e.g., meetings to review the English versions, proof readers, etc. Additionally, it sounds as if many languages were sight-translated from an English version of the survey? How were those team members trained to ensure comprehension of the meaning and intention of the English survey items? How did they communicate mental health vocab or other challenging vocab where linguistic and community equivalent terms may be harder to find? Did they develop a glossary for key terms?

Response: PANA staff initially translated the survey into Somali and Arabic through outside consultants. PANA staff engaged in two or three training sessions discussing the survey questions with the primary researchers to understand what words mean. Because staff come from the communities we serve and speak the language, they were able to conduct the interviews and help folks understand what we mean.

5. Is there a reason the authors elected not to share data on differences between language groups or include language in the adjusted model? I realize this is a nuanced decision. But communities who speak languages of lesser diffusion may face enhanced barriers to Section 8 applications.

Response: We agree with this observation. The decision was a practical one driven primarily by statistical necessity. We conducted qualitative work concurrent with the community survey which allowed us to explore these topics with more nuance (see: https://www.panasd.org/refugee-experiences-report). Because language and region of origin maps on to historic waves of refugee resettlement in San Diego, which in turn correlates with specific areas of the county in which individuals settled, primary language group is also significantly confounded by year of arrival and acculturation. For the purposes of this manuscript, we therefore chose to focus on experiences that were common across the refugee community, in concert with PANA’s mission to build community power through solidarity.

Reviewer #2: 

This manuscript is on an important topic for refugee health. The analysis is meaningful and would be an important contribution to the literature. However, the manuscript needs substantial revisions throughout. Below please find some feedback.

Response: Thank you for your feedback. Your suggestions have been noted and have strengthened this manuscript. 

Title

o I suggest using more of an objective word like low instead of the subjective word poor in the title and throughout.

Response: We have replaced the word “poor” with the word “low.” Thank you for the suggestion.

o I suggest adding the time period in the title as outcomes change throughout time.

Response: We have added “from September to November of 2020” to the title.

Abstract

o The total sample size is 544, but it reads as half the sample size in the abstract. I suggest re-writing this to make it clearer.

Response: We have re-wrote the abstract to match the correct sample size.

o I suggest replacing “fewer” odds with lower odds.

Response: We have replaced “fewer” with “lower” odds.

Introduction

o I suggest adding bolded subheadings throughout the introduction to guide the reader.

Response: We have added bolded subheadings throughout the introduction.

o A reference is needed for the first sentence.

Response: We have added a reference to the first sentence.

o There are a couple of run-on sentences at the end of the first paragraph.

Response: We have reformatted the sentences so that they flow better and are no longer run-on sentences.

o Some references are numbered and some are authors last name.

Response: We have properly reformatted the references throughout the manuscript.

o Period missing at the end of the introduction section.

Response: We have added the missing period. Thank you for catching that error!

Methods

o The STROBE guidelines should be used to make the methods section clear and consistent with previous literature. Sections should include setting, participants, variables, data source, and statistical methods. https://www.equator-network.org/wp-content/uploads/2015/10/STROBE_checklist_v4_combined.pdf

Response: We have incorporated relevant STROBE guidelines to restructure the Methods section.

o “Interviews were limited to no more than two individuals per household (typically a parent and an adult child).” – How many interviews were completed with individuals from the same household? This should be mentioned as a limitation in the discussion.

Response: Our partners who conducted the interviews kept track of who was interviewed. However, we did not record how many households had more than one participant in the study. We have added this to the limitations section. 

o How many questions were on the survey? About how long did it take for participants to complete?

Response: The survey contained 83 questions regarding demographics, COVID-19, housing, employment, health, children, belonging, and resilience. The survey took approximately 30 minutes to complete and up to 1 hour if it was being translated.

o The statistical methods are mentioned before describing the survey and variables, which should come first.

Response: We have moved the survey/variable descriptions before the statistical methods section.

o “Data gathered from the questionnaires was anonymous” – How do you know if a participants completed the survey more than once? This could be mentioned in the limitations.

Response: “PANA staff worked from a list of families. Each staff interviewer was assigned a list of people based on language. Staff interviewed and conducted the survey by phone with each individual ensuring no duplication.” We added a couple sentences to clarify in the methods section under participants. 

o Data is based on self-report, which should be described in the limitations.

Response: We have added this to the limitations section.

o What options were given for gender? Were there responses available for: prefer not to respond, non-binary, or transgender?

Response: There were only options given for male and female to preserve statistical significance. 

Results

o The first paragraph of the results jumps into a comparison without first describing the overall population of respondents. How old were the respondents for example? What regions or countries are respondents from? What percent of respondents are refugees versus asylum seekers? How many of the respondents are refugees/asylees versus children of refugees/asylees born in the US?

Response: We have addressed each of these questions in Table 1. However, we did not record which respondents were refugees/asylees versus children of refugees/asylees born in the US.

o An overall demographics table would be valuable as an initial table.

Response: We have created a demographics table (Table 1).

o This sentence describes methods and should be moved to the methods section: “In consultation with PANA leadership, the study team decided to dichotomize this variable and compare participants who reported ‘never’ experiencing emotional problems vs all others.”

Response: We have moved this sentence to the location you suggested.

o This sentence interprets the results and should be moved to the discussion section: “Discussing mental health is highly stigmatized in the community, and the study team agreed that disclosing any concerns related to emotional or mental health in this context was likely indicative of significant levels of emotional distress.”

Response: We have removed this interpretive sentence from the results section.

Discussion

I suggest adding bolded subheadings throughout the discussion to guide the reader.

Response: We have added bolded subheadings throughout the discussion.

 I suggest re-wording the findings to describe how more support is needed for this population as opposed to it reading as if this population is not taking care of their health and putting others at risk.

Response: Thank you for this suggestion. We have reworded the findings so as to not suggest the risks faced by this group are because of their choices but due to structural issues.

The second paragraph of the discussion should be moved to a limitations section at the end of the discussion before the conclusions.

Response: We have revised the discussion so that this section has been moved to the limitations section prior to the conclusion.

o More of a description of the time period is needed related to COVID-19 and testing. For example, how accessible were COVID-19 testing at that time? When did the government send out home tests for free?

Response: This study was conducted during the Fall of 2020. Testing was widely available at clinics and testing centers but antigen “home tests” were not yet available in meaningful amounts. It was not until the first year of the Biden Administration that the government had the capacity to send home tests to households.

o This sentence implies causality instead of an association: “housing conditions can add an extra emotional and mental burden”

Response: We have modified this sentence so that it does not imply causality.

o United States and US are used. I suggest writing out in full each time.

Response: We have replaced “US” with “United States”.

o “San Diegan” – Perhaps re-word to refugees living in San Diego.

Response: We have replaced “San Diegan refugees” with “refugees living in San Diego.”

o A limitations section was missing.

Response: We have added a limitation section and have added subheaders.

o It would be great to see more specifics about next steps related to affordable housing mentioned in the conclusions. What are some specific steps that could be taken to address this issue? Have other counties, states, or countries successfully found affordable housing solutions?

Response: Thank you for your comment, we have added the following statement in our conclusion: 

In addition, we recommend the federal government increase the number of Section 8 vouchers available so that wait times are reduced (34). Finally, we recommend that the State of California create legislation and provide funding for permanent social housing that refugee populations can access (10).

These policy recommendations and more can be found in the PANA report. 

Tables and Figures

Tables should include more comprehensive titles that answer who (refugee and asylum seekers), what, where (San Diego County, CA), and when (time period).

Response: We have included comprehensive titles to our tables as you have suggested.

I suggest consistency in the number of significant digits in the p values in the tables.

Response: We have standardized the number of significant digits in the table p-values.

Reviewer #3: 

A very nice community-engaged applied research study addressing an extremely important topic. 

Response: Thank you for this comment. 

Suggestions: 

I suggest that they re-arrange Table 1 so that they report measures in the table in the same order in which they report them in the text.

Response: We added a new demographics table (Now Table 1) and have rearranged the table you were originally referencing (Now Table 2) to report measures in the same order in which we report them in the text

---

## [Decision Letter · Decision Letter 1]

19 Dec 2022

PONE-D-22-20522R1Overcrowded Housing Reduces COVID-19 Mitigation Measures and Lowers Emotional Health Among San Diego Refugees from September to November of 2020PLOS ONE

Dear Dr. Hassani,

Thank you for submitting your manuscript to PLOS ONE. After careful consideration, we feel that it has merit but does not fully meet PLOS ONE’s publication criteria as it currently stands. Therefore, we invite you to submit a revised version of the manuscript that addresses the points raised during the review process.

ACADEMIC EDITOR: This study is on a very important topic for public health. However, the authors are required to make corrections by addressing the reviewers criticisms. Criticisms of methods and findings should be taken seriously. Although the authors report that they have used the STROBe guideline, there are missing information.==============================

We look forward to receiving your revised manuscript.

Kind regards,

Ugurcan Sayili, M.D.

Academic Editor

PLOS ONE

Reviewers' comments:

Reviewer's Responses to Questions

**Comments to the Author**

1. If the authors have adequately addressed your comments raised in a previous round of review and you feel that this manuscript is now acceptable for publication, you may indicate that here to bypass the “Comments to the Author” section, enter your conflict of interest statement in the “Confidential to Editor” section, and submit your "Accept" recommendation.

Reviewer #2: (No Response)

Reviewer #4: All comments have been addressed

2. Is the manuscript technically sound, and do the data support the conclusions?

Reviewer #2: Partly

Reviewer #4: Yes

3. Has the statistical analysis been performed appropriately and rigorously? 

Reviewer #2: Yes

Reviewer #4: No

4. Have the authors made all data underlying the findings in their manuscript fully available?

Reviewer #2: No

Reviewer #4: No

5. Is the manuscript presented in an intelligible fashion and written in standard English?

Reviewer #2: Yes

Reviewer #4: Yes

6. Review Comments to the Author

Reviewer #2: This manuscript is on an important topic for refugee health. The analysis is meaningful and would be an important contribution to the literature. Below please find some feedback.

Abstract

o I suggest deleting the word “extremely.”

o Grammatical errors are included throughout the abstract including missing preposition and space.

o “Odds” should be used not “likely” when using odds ratios.

Introduction

o I suggest adding descriptive bolded subheadings throughout the introduction to guide the reader besides background.

o I suggest a more specific objectives section that answers who, what, where, and when.

Methods

o The study was a cross-sectional study design, which was missing from the study design section.

o This seems to fit better under the participant section: “Individuals were eligible

to participate in the study if they were over the age of 18, able to participate in a language spoken by a trained research assistant, willing to provide informed consent, and part of the refugee community. “Refugee community member” was construed broadly and included individuals who had arrived as refugees or asylum seekers or their American born children.”

Results

o Table 1 figure title was missing who (the population), where (CA), and when (timeframe). I suggest writing out Southeast and East.

o “Odds” should be used not “likely” when using odds ratios.

o Table 2 figure title – I suggest including refugee participants to the title. I suggest using the same number of significant digits in the p values. I usually see n (%) not % (n) in tables, which you may considering changing.

o Table 3 – Perhaps add an asterisk for the odds ratios that are statistically significant.

Discussion

o I suggest adding more descriptive bolded subheadings throughout the discussion to guide the reader.

o “Odds” should be used throughout the discussion not “likely” when using odds ratios as the methods.

o “There is a strong effect size” – This language does not match the statistical methodology

o I suggest deleting “extremely” in extremely high rates of job loss.

o I suggest adding only two gender responses as a limitation.

o The conclusions section is fairly repetitive of the discussion. I suggest adding more specific future directions in this section.

Reviewer #4: Thank the researchers for making the necessary corrections to the suggestions of the referees.

We have a few more suggestions for the development of this well-written study that will benefit the literature.

1. Some abbreviations are not mentioned in the article (HUD, E, SE)

2. The type of study that the research belongs to should be stated in the method section.

3. Although the researchers state that they follow the STROBE guideline, the article does not contain how the sample size is calculated. In addition, since researchers make use of statistical analysis methods, they should specify the statistical analysis methods they use in the method section by creating a separate section.

4. In Table 1, n's should be written in small letter. N is the frequency in the population, n is the frequency in the sample.

5. While the researchers stated that they only used simple logistic regression analysis, Table 2 contains different statistical analyzes (Chi-square, Mann Whitney U?). The tests used should be indicated by marking the relevant p values in the method section and under the Table. The normality tests or methods of continuous data and descriptive statistics should also be specified in the method.

6. In Table 1, continuous data are indicated as median (range), while in Table 2 they are indicated as mean (SD). Researchers should indicate how they changed the descriptives.

7. Age and Years in the United States are considered to be continuous data in Table 1 but are shown as % (n). This confusion must be cleared.

8. In Table 3, under the table, which variables are added to the multivariate model should be stated and the R2 value should be added.

9. If the multivariable logistic regression model is used, the method by which it is used (Backward, Enter?), the tests that examine the suitability of the model should also be added to the method.

10. The researchers stated that there were 83 questions in the questionnaire, but not all variables were included in the regression analysis. The criteria for inclusion in the model should be specified.

11. The reference categories of categorical variables should be indicated in Table 3.

12. It is not understood how the researchers created the regression model, is it possible to reach the COVID-19 test as a result of living in a crowded house, or is it possible to live in a crowded house because there is no access to the COVID-19 test? Although it is thought that the researchers associated this with low income and therefore it was added to the model, it would be good to state this situation in the text and add it to the discussion with a few words.

13. Lower emotional health is not understood as a predictive factor for overcrowded housing. It seems more logical to us that it occurs as a result of overcrowded housing. Researchers should explain why they add it to the model and how they see it as a predictive factor (Researchers stated in the method section “We conducted a simple logistic regression to examine the predictors for overcrowded housing.”). While it was mentioned in the discussion, "There is evidence that overcrowded housing can have a negative impact on an individual's mental health (27,28). Our findings add to this literature but our narrowed focus on the refugee community highlights that housing conditions can add an extra emotional and mental burden to a population already at high risk of lower emotional/mental health outcomes (29.30).” . However, the analysis is a regression analysis of the effect of low emotional health on overcrowded housing.

7. PLOS authors have the option to publish the peer review history of their article (what does this mean?). If published, this will include your full peer review and any attached files.

Reviewer #2: No

Reviewer #4: **Yes: **Betül Zehra Pirdal

---

## [Author Response · Author response to Decision Letter 1]

25 Jan 2023

Thank you all for your feedback and suggestions.

Reviewer #2

Abstract

o I suggest deleting the word “extremely.”

Response: We have removed the word “extremely” from the abstract.

o Grammatical errors are included throughout the abstract including missing prepositions and space.

Response: We have changed all identifiable missing spaces and prepositions.

o “Odds” should be used not “likely” when using odds ratios.

Response: We have removed the word “likely” when discussing odds ratios in the Abstract. Please see highlighted corrections in the manuscript.

Introduction

o I suggest adding descriptive bolded subheadings throughout the introduction to guide the reader besides the background.

Response: We defer to the editor for the preferred format of the manuscript.

o I suggest a more specific objectives section that answers who, what, where, and when.

Response: We believe that delving into too much detail would be redundant in this section.

Methods

o The study was a cross-sectional study design, which was missing from the study design section.

Response: We now have mentioned that this is a cross-sectional study in the study design section.

o This seems to fit better under the participant section: “Individuals were eligible

to participate in the study if they were over the age of 18, able to participate in a language spoken by a trained research assistant, willing to provide informed consent, and part of the refugee community. “Refugee community member” was construed broadly and included individuals who had arrived as refugees or asylum seekers or their American-born children.”

Response: We have moved the following section to the “participants” section. To reduce redundancy we also removed the following sentence from the participant section: “To be eligible to participate in the survey individuals had to be 18 years old or above and identify as a refugee, asylum seeker, or the child of a refugee or asylum seeker.”

Results

o Table 1 figure title was missing who (the population), where (CA), and when (timeframe). I suggest writing out Southeast and East.

Response: We have adjusted the table title to: “Sample Demographics of the refugee participants in San Diego, CA, sampled between November 2020-December 2020. Includes age, average years in the United States, English spoken at home, and gender stratified by region of origin.”

o “Odds” should be used not “likely” when using odds ratios.

Response: We have removed the word “likely” when discussing odds ratios in the manuscript. Please see highlighted corrections in the manuscript.

o Table 2 figure title – I suggest including refugee participants to the title. I suggest using the same number of significant digits in the p values. I usually see n (%) not % (n) in tables, which you may consider changing.

Response: We have adjusted the title. We have also switched the “n” and “%” in the table. The p values sig figs have also been addressed

o Table 3 – Perhaps add an asterisk for the odds ratios that are statistically significant.

Response: We have added asterisks where there is statistical significance. Thank you for your suggestion. 

Discussion

o I suggest adding more descriptive bolded subheadings throughout the discussion to guide the reader.

Response: We defer to the editor for the preferred format of the manuscript.

o “Odds” should be used throughout the discussion not “likely” when using odds ratios as the methods.

Response: We have removed the word “likely” when discussing odds ratios in the discussion. Please see highlighted corrections in the manuscript.

o “There is a strong effect size” – This language does not match the statistical methodology

Response: We have replaced the word “strong effect size” with “significant association” to reflect our statistical methodology. 

o I suggest deleting “extremely” in extremely high rates of job loss.

Response: We have now deleted all mentions of the word “extremely.”

o I suggest adding only two gender responses as a limitation.

Response: We have added the following statement in the limitations section: “Additionally, gender was limited to two options (male or female) to preserve statistical significance and we are aware that this may not be a true representation of the diverse population.”

o The conclusions section is fairly repetitive in the discussion. I suggest adding more specific future directions in this section.

Response: We have added a specific future direction for research: “Future directions in research should look at the impact of master leasing, housing vouchers, and universal basic income on refugee housing security and general health outcomes.”

Reviewer #4

Reviewer #4: Thank the researchers for making the necessary corrections to the suggestions of the referees.

We have a few more suggestions for the development of this well-written study that will benefit the literature.

1. Some abbreviations are not mentioned in the article (HUD, E, SE)

Response: Thank you for catching that. We have removed these abbreviations and fully written them out.

2. The type of study that the research belongs to should be stated in the method section.

Response: We have defined this study as a cross-sectional study and have now added that term to the study design section of the methods.

3. Although the researchers state that they follow the STROBE guideline, the article does not contain how the sample size is calculated. In addition, since researchers make use of statistical analysis methods, they should specify the statistical analysis methods they use in the method section by creating a separate section.

Response: The following description of sample size considerations is now included in the methods section:

Data were collected as part of a programmatic report intended to provide a general overview of the community. As such, we did not conduct a priori power calculations focused on one specific outcome.

Per recommendations from the STROBE guidelines, we have elected to use confidence intervals to contextualize the precision of our results, rather than conducting post-hoc power calculations(1,2).

4. In Table 1, n's should be written in small letters. N is the frequency in the population, and n is the frequency in the sample.

Response: Thank you, Table 1 has been edited to reflect the suggested changes.

5. While the researchers stated that they only used simple logistic regression analysis, Table 2 contains different statistical analyzes (Chi-square, Mann Whitney U?). The tests used should be indicated by marking the relevant p values in the method section and under the Table. The normality tests or methods of continuous data and descriptive statistics should also be specified in the method.

Response: We appreciate this point, we have added the following text to the methods for clarity:

 “We first examined basic univariate frequencies as a study team to develop a basic understanding of sample demographics and the prevalence of indicators of interest within the sample. We then used chi-square and student’s t-tests to test the hypothesis that our primary predictor and covariates of interest were significantly associated with living in severely overcrowded housing.”

6. In Table 1, continuous data are indicated as median (range), while in Table 2 they are indicated as mean (SD). Researchers should indicate how they changed the descriptives.

Response: Thank you. Table 1 has been amended to be more consistent with Table 2.

7. Age and Years in the United States are considered to be continuous data in Table 1 but are shown as % (n). This confusion must be cleared.

Response: Thank you for the input, we have edited the table to be less confusing. 

8. In Table 3, under the table, which variables are added to the multivariate model should be stated and the R2 value should be added.

Response: We have added more detail regarding model fit statistics:

“The fully adjusted model is shown in Table 3. The model had a relatively good fit, with a pseudo-r-squared value of 0.20, no specification errors identified using the linktest command in Stata (_hatsq p = 0.68), a non-significant Hosmer-Lemeshow goodness-of-fit test (p=0.78).”

9. If the multivariable logistic regression model is used, the method by which it is used (Backward, Enter?), and the tests that examine the suitability of the model should also be added to the method.

Response: Variables were chosen based on theoretical considerations, per Aneshensel’s recommendations (3), and in dialogue with the leadership of the community organization which led the data collection efforts. Our team decided that utilizing step-wise regressions for model design would be inappropriate for a model utilizing a hypothesis and theory-driven approach.

10. The researchers stated that there were 83 questions in the questionnaire, but not all variables were included in the regression analysis. The criteria for inclusion in the model should be specified.

Response: Variables were chosen based on the specific associations of interest (crowded housing and mental health, crowded housing, and testing behaviors). Additional covariates were added to address potential issues of confounding. We have clarified this by adding the following language to the methods section:

“Variables for the present analyses were chosen in consultation with PANA leadership to address the primary research question: What is the association between housing and COVID-19 vulnerability? Vulnerability was broadly understood as both healthcare access and the broader mental health impact of the epidemic. The full report with all variables can be found elsewhere.”

11. The reference categories of categorical variables should be indicated in Table 3.

Response: As the majority of variables in Table 3 are binary we elected not to include the reference variable in order to be parsimonious (i.e., “Never accessed covid test” vs “Ever accessed covid test”). While the emotional health variable is ordinal (almost always/always/most of the time/about half the time/never”) we felt it was conceptually reasonable to treat it as a continuous variable given that the actual distribution of emotional well-being would follow this distribution. We have clarified the latter point in the methods section with the following language:

 We built simple logistic regression models to measure the unadjusted odds of association with overcrowded housing and finally constructed a full multivariate regression with all variables of interest to measure the adjusted odds of association. While the emotional health item was collected using a Likert scale, it was modeled as a continuous variable as this was the most conceptually consistent way to consider an individual’s emotional wellbeing. 

12. It is not understood how the researchers created the regression model, is it possible to reach the COVID-19 test as a result of living in a crowded house, or is it possible to live in a crowded house because there is no access to the COVID-19 test? Although it is thought that the researchers associated this with low income and therefore it was added to the model, it would be good to state this situation in the text and add it to the discussion with a few words.

Response: We agree that the results are interesting, which is why we are excited to share them with a broader academic audience to spark discussion. In the discussion, we suggest one possible mechanism:

“If a person is living in severely overcrowded housing without the ability or resources to properly quarantine or isolate then the results of a COVID-19 test may not have much impact on behavior, which can diminish incentives for testing.”

13. Lower emotional health is not understood as a predictive factor for overcrowded housing. It seems more logical to us that it occurs as a result of overcrowded housing. Researchers should explain why they add it to the model and how they see it as a predictive factor (Researchers stated in the method section “We conducted a simple logistic regression to examine the predictors for overcrowded housing.”). While it was mentioned in the discussion, "There is evidence that overcrowded housing can have a negative impact on an individual's mental health (27,28). Our findings add to this literature but our narrowed focus on the refugee community highlights that housing conditions can add an extra emotional and mental burden to a population already at high risk of lower emotional/mental health outcomes (29.30).” However, the analysis is a regression analysis of the effect of low emotional health on overcrowded housing.

Thank you for bringing this up. We have updated the title of the table to make it less confusing: “Simple and adjusted odds of living in severely overcrowded housing vs. not living in severely overcrowded housing.”Additionally, we have revised the language in the manuscript and replaced the word predictor with association to prevent confusion. 

We have addressed this somewhat in the methods section. Our decision to consider mental health was based on previous research conducted by our team. The decision was also based on consultation with the PANA leadership team, who pointed out that living in substandard (i.e., severely overcrowded) housing can be a source of emotional distress and anxiety for many individuals.

References

Vandenbroucke, J. P., von Elm, E., Altman, D. G., Gøtzsche, P. C., Mulrow, C. D., Pocock, S. J., Poole, C., Schlesselman, J. J., Egger, M., & STROBE Initiative (2007). Strengthening the Reporting of Observational Studies in Epidemiology (STROBE): explanation and elaboration. PLoS medicine, 4(10), e297. https://doi.org/10.1371/journal.pmed.0040297

Cuschieri S. (2019). The STROBE guidelines. Saudi journal of anaesthesia, 13(Suppl 1), S31–S34. https://doi.org/10.4103/sja.SJA_543_18

Aneshensel, C. S. (2012). Theory-based data analysis for the social sciences. Sage Publications.

---

## [Decision Letter · Decision Letter 2]

20 Feb 2023

PONE-D-22-20522R2Overcrowded Housing Reduces COVID-19 Mitigation Measures and Lowers Emotional Health Among San Diego Refugees from September to November of 2020PLOS ONE

Dear Dr. Hassani,

Thank you for submitting your manuscript to PLOS ONE. After careful consideration, we feel that it has merit but does not fully meet PLOS ONE’s publication criteria as it currently stands. Therefore, we invite you to submit a revised version of the manuscript that addresses the points raised during the review process.

ACADEMIC EDITOR: The article is on an important topic, and I think it can be published with corrections. However, I agree with reviewer 4 indicated the problem on regression analysis. Unless the authors properly perform the regression analysis, the manuscript will not be ready for publication. Please choose the dependent and independent variables appropriately.

We look forward to receiving your revised manuscript.

Kind regards,

Ugurcan Sayili, M.D.

Academic Editor

PLOS ONE

Reviewers' comments:

Reviewer's Responses to Questions

**Comments to the Author**

1. If the authors have adequately addressed your comments raised in a previous round of review and you feel that this manuscript is now acceptable for publication, you may indicate that here to bypass the “Comments to the Author” section, enter your conflict of interest statement in the “Confidential to Editor” section, and submit your "Accept" recommendation.

Reviewer #2: All comments have been addressed

Reviewer #3: All comments have been addressed

Reviewer #4: All comments have been addressed

2. Is the manuscript technically sound, and do the data support the conclusions?

Reviewer #2: Yes

Reviewer #3: Yes

Reviewer #4: Partly

3. Has the statistical analysis been performed appropriately and rigorously? 

Reviewer #2: Yes

Reviewer #3: Yes

Reviewer #4: No

4. Have the authors made all data underlying the findings in their manuscript fully available?

Reviewer #2: No

Reviewer #3: Yes

Reviewer #4: No

5. Is the manuscript presented in an intelligible fashion and written in standard English?

Reviewer #2: Yes

Reviewer #3: Yes

Reviewer #4: Yes

6. Review Comments to the Author

Reviewer #2: This manuscript is on an important topic for refugee health. The analysis is meaningful and would be an important contribution to the literature. Below please find some feedback.

• Abstract

o The copied and pasted abstract within the submission portal section near key words did not include the same edits as the abstract within the revised manuscript.

o Rather than saying more than twice the odds and nearly 4 times the odds, use the exact number for the odds. Ex: 3.9 times the odds.

• Introduction

o I suggest a more specific objectives section that answers who, what, where, and when. The current objectives are more like a study purpose.

• Methods

o “able to participate in a language” Perhaps change to speaks one of the languages included in the study

o Incomplete sentence: “Demographic variables such as income from the previous week (Have you worked for money in the previous week), age, year of arrival

to the United States, gender (male or female options only), cohabiting with a partner, number of children (How many children under the age of 18 are you responsible for?), and family size.”

• Was your data normally distributed? This should be checked before running a t test.

• Was it a multivariate regression or multiple regression?

• Results

o Table 1 figure title – I suggest deleting “Include age, average years in the US, etc

o I suggest rephrasing: “were 81% lower odds of reporting living in severely crowded conditions?”

• Discussion

o The conclusions are repetitive of findings and don’t instead focus more on the future

Reviewer #3: Thank you for your revisions. No further comments.

Reviewer #4: Researchers included “access to the COVID-19 test” and “Lower emotional health” as independent factors in the regression model affecting “severely overcrowded housing”, the outcome variable of the article. The researchers created the inverse relationship between dependent and independent factors, and although this inverse relationship was mentioned in the first review, no changes were made in the analysis, only the conclusion sentences were changed. Since the regression analysis was not created appropriately in the article, it is not suitable for publication.

7. PLOS authors have the option to publish the peer review history of their article (what does this mean?). If published, this will include your full peer review and any attached files.

Reviewer #2: No

Reviewer #3: No

Reviewer #4: No

---

## [Author Response · Author response to Decision Letter 2]

20 Apr 2023

PLOS ONE - Review

February 10, 2023

Title: Overcrowded Housing Reduces COVID-19 Mitigation Measures and Lowers

Emotional Health Among San Diego Refugees from September to November of 2020

This manuscript is on an important topic for refugee health. The analysis is meaningful and would be an important contribution to the literature. Below please find some feedback.

· Abstract

o The copied and pasted abstract within the submission portal section near key words did not include the same edits as the abstract within the revised manuscript.

Response: Thank you for catching that, we will update in the author’s portal accordingly. 

o Rather than saying more than twice the odds and nearly 4 times the odds, use the exact number for the odds. Ex: 3.9 times the odds.

 Response: Based on comments from another reviewer we have restructured our analyses and consequently our results. We have elected to describe the findings more qualitatively in the abstract to ensure that the holistic meaning of the findings is highlighted, rather than the somewhat complex interpretations necessary for standardized path coefficients. 

· Introduction

o I suggest a more specific objectives section that answers who, what, where, and when. The current objectives are more like a study purpose.

Response: We appreciate the reviewer's stylistic suggestion. We have opted to retain the original language to conserve space and because we feel that it is important to highlight the overall purpose of the study in the background before specifying the details the reviewer emphasizes (who, what, where, when) at more length in the methods section. 

· Methods

o “able to participate in a language” Perhaps change to speaks one of the languages included in the study

Response: We have changed this language to: “could speak one of the languages included in the study”

o Incomplete sentence: “Demographic variables such as income from the previous week (Have you worked for money in the previous week), age, year of arrival to the United States, gender (male or female options only), cohabiting with a partner, number of children (How many children under the age of 18 are you responsible for?), and family size.”

Response: We have adjusted this incomplete sentence to: “We also included demographic variables such as income from the previous week (Have you worked for money in the previous week), age, year of arrival to the United States, gender (male or female options only), cohabiting with a partner, number of children (How many children under the age of 18 are you responsible for?), and family size.”

· Was your data normally distributed? This should be checked before running a t test.

Response: Yes, it was normally distributed. Moreover, structural equation model global fit can be evaluated based using tests which do and do not assume normality and so are somewhat robust to these assumptions.

· Was it a multivariate regression or multiple regression?

Response: We have updated our analysis based on comments made by reviewer #4.

· Results

o Table 1 figure title – I suggest deleting “Include age, average years in the US, etc

Response: We have removed this sentence from the Table 1 title.

o I suggest rephrasing: “were 81% lower odds of reporting living in severely crowded conditions?”

Response: We have updated our analysis based on comments made by reviewer 4 that have made this suggestion no longer applicable. 

· Discussion

o The conclusions are repetitive of findings and don’t instead focus more on the future

Response: Thank you. We have removed language that repeats findings and have added more language on recommendations and future directions.

o Reviewer #4: Researchers included “access to the COVID-19 test” and “Lower emotional health” as independent factors in the regression model affecting “severely overcrowded housing”, the outcome variable of the article. The researchers created the inverse relationship between dependent and independent factors, and although this inverse relationship was mentioned in the first review, no changes were made in the analysis, only the conclusion sentences were changed. Since the regression analysis was not created appropriately in the article, it is not suitable for publication.

Response: We concur that originally, it was confusing to follow the relationships between dependent and independent factors. We have since revised the analysis utilizing a structural equation modeling approach which more heavily emphasizes a priori theories of association (see table 3 and figure 1 in revised manuscript). We also accordingly updated our results section and abstract. 

References:

 1. Kline, R. B. (1998). Structural equation modeling. New York: Guilford.

---

## [Decision Letter · Decision Letter 3]

29 May 2023

Overcrowded Housing Reduces COVID-19 Mitigation Measures and Lowers Emotional Health Among San Diego Refugees from September to November of 2020

PONE-D-22-20522R3

Dear Dr. Hassani,

We’re pleased to inform you that your manuscript has been judged scientifically suitable for publication and will be formally accepted for publication once it meets all outstanding technical requirements.

Kind regards,

Ugurcan Sayili, M.D.

Academic Editor

PLOS ONE

Additional Editor Comments (optional):

Reviewers' comments:

Reviewer's Responses to Questions

**Comments to the Author**

1. If the authors have adequately addressed your comments raised in a previous round of review and you feel that this manuscript is now acceptable for publication, you may indicate that here to bypass the “Comments to the Author” section, enter your conflict of interest statement in the “Confidential to Editor” section, and submit your "Accept" recommendation.

Reviewer #2: All comments have been addressed

2. Is the manuscript technically sound, and do the data support the conclusions?

Reviewer #2: Yes

3. Has the statistical analysis been performed appropriately and rigorously? 

Reviewer #2: Yes

4. Have the authors made all data underlying the findings in their manuscript fully available?

Reviewer #2: (No Response)

5. Is the manuscript presented in an intelligible fashion and written in standard English?

Reviewer #2: Yes

6. Review Comments to the Author

Reviewer #2: This manuscript is on an important topic for refugee health. The analysis is meaningful and would be an important contribution to the literature. The previous suggested revisions have been incorporated into this submission and have improved the manuscript. Below please find some minor feedback related to the results.

• “Just over 1 in 4 participants reported that they had engaged in work for money in the previous two weeks.” I suggest removing these estimations from the results and using exact numbers.

• Table 2 title – I suggest removing “overcrowded vs.” from the title as the comparison is between not severely overcrowded and severely overcrowded

• Table 2 – I suggest adding female to the top above age; I suggest moving the emotional problems question to the bottom of the table

• Table 3 – Define acronyms below the table; Shorten the title and put the description below the table; I suggest consistency in the significant digits of the p values

7. PLOS authors have the option to publish the peer review history of their article (what does this mean?). If published, this will include your full peer review and any attached files.

Reviewer #2: No

---

## [Editor Report · Acceptance letter]

8 Jun 2023

PONE-D-22-20522R3 

Overcrowded Housing Reduces COVID-19 Mitigation Measures and Lowers Emotional Health Among San Diego Refugees from September to November of 2020 

Dear Dr. Hassani:

I'm pleased to inform you that your manuscript has been deemed suitable for publication in PLOS ONE. Congratulations! Your manuscript is now with our production department. 

Kind regards, 

on behalf of

Dr. Ugurcan Sayili 

Academic Editor

PLOS ONE